# Highly Efficient Vertical Transmission for Zika Virus in *Aedes aegypti* after Long Extrinsic Incubation Time

**DOI:** 10.3390/pathogens9050366

**Published:** 2020-05-11

**Authors:** Menchie Manuel, Dorothée Missé, Julien Pompon

**Affiliations:** 1Department of Emerging Infectious Diseases, Duke-NUS Medical School, Singapore 169857, Singapore; menchie.manuel@duke-nus.edu.sg; 2CNRS, IRD, MIVEGEC, Univ. Montpellier, 34394 Montpellier, France; dorothee.misse@ird.fr

**Keywords:** Zika virus, *Aedes aegypti*, vertical transmission

## Abstract

While the Zika virus (ZIKV) 2014–2017 pandemic has subsided, there remains active transmission. Apart from horizontal transmission to humans, the main vector *Aedes aegypti* can transmit the virus vertically from mother to offspring. Large variation in vertical transmission (VT) efficiency between studies indicates the influence of parameters, which remain to be characterized. To determine the roles of extrinsic incubation time and gonotrophic cycle, we deployed an experimental design that quantifies ZIKV in individual progeny and larvae. We observed an early infection of ovaries that exponentially progressed. We quantified VT rate, filial infection rate, and viral load per infected larvae at 10 days post oral infection (d.p.i.) on the second gonotrophic cycle and at 17 d.p.i. on the second and third gonotrophic cycle. As compared to previous reports that studied pooled samples, we detected a relatively high VT efficiency from 1.79% at 10 d.p.i. and second gonotrophic cycle to 66% at 17 d.p.i. and second gonotrophic cycle. At 17 d.p.i., viral load largely varied and averaged around 800 genomic RNA (gRNA) copies. Longer incubation time and fewer gonotrophic cycles promoted VT. These results shed light on the mechanism of VT, how environmental conditions favor VT, and whether VT can maintain ZIKV circulation.

## 1. Introduction

Zika virus (ZIKV) is a mosquito-transmitted *Flavivirus* that recently emerged as a pandemic virus [1]. From 2014–2017, ZIKV infected more than 1.5 million people mostly in Polynesia and Latin and South America. Although Zika symptoms are mild (flu-like) to absent, it can have life-debilitating and life-threatening consequences, such as microcephaly for prenatally infected newborns and Guillain–Barre syndrome in adults [2]. The wide distribution of and health risks associated with the infection prompted the World Health Organization (WHO) to declare a public health emergency of international concern in 2016. However, after 2018, records of Zika cases sharply dropped in the Americas and Polynesia [3] and the emergency status was lifted. The collapse of the epidemic was likely caused by herd immunity, given that almost 60% of the population had immunity against ZIKV in the Americas and Polynesia [4,5]. Nonetheless, Zika cases are still reported in Asia and suspected in Africa [3], and autochthonous transmission in France occurred at the end of 2019 [6]. These reports indicate that ZIKV circulation persists. A better understanding of ZIKV ongoing circulation will help prevent future outbreaks. Identification of the mode of transmission to target will help curb residual circulation.

While ZIKV is transmitted horizontally from mosquito to human, it is also vertically transmitted from mother to offspring in mosquitoes as for many pathogenic flaviviruses [7,8]. Moreover, ZIKV can be transmitted between humans during sexual intercourse and by maternal–fetal infection [9]. *Aedes aegypti* is the most likely horizontal vector of ZIKV, although *Aedes albopictus* is also competent to a lower extent [9,10,11]. Evidence of mosquito vertical transmission (VT) comes from laboratory studies that evaluate the ability of the virus to be passed onto the next generation and from detection of the virus in immature stages in the field. In controlled conditions, following oral infection, which is more relevant than intrathoracic inoculation, ZIKV VT has repeatedly been shown for *Ae. aegypti* [12,13,14]. ZIKV was also repeatedly detected in eggs and larvae collected in urban and forested environments [15,16,17,18]. However, a few other studies report an absence of VT in the laboratory or lack of detection in immature stages in the field [19,20]. Altogether, there is enough evidence to support the existence of VT for ZIKV in *Ae. aegypti*. However, VT efficiency varies with several parameters including mosquito and virus genetics, temperature, gonotrophic cycle, and extrinsic incubation period [7]. Variation in these parameters may explain the lack of VT experimental observation and absence of detected VT in the field. To understand how ZIKV circulation persists and especially how horizontal and vertical transmissions are balanced, it is important to determine the conditions that promote ZIKV VT.

In arthropods, two mechanisms can result in VT [21]. Infection of eggs when fertilized or oviposited can occur in the oviduct and is called trans-ovum transmission. Alternatively, infection of the female germinal tissues results in infected progeny and is called transovarial transmission. Both mechanisms have been proposed for arbovirus VT in *Ae. aegypti* [22,23]. Following uptake of an infectious blood meal, the viruses first infect the mosquito midgut and then disseminate to other tissues including muscles, legs, wings, trachea, and salivary glands [24]. ZIKV salivary gland infection can occur as early as three days post infection, leading to virus secretion in saliva and horizontal transmission to another host [10]. The whole mosquito body can virtually be infected by arboviruses at late incubation time. There is evidence of virus infection of ovaries and oviduct [25], supporting both trans-ovum and transovarial mechanisms. Importantly, the impact of gonotrophic cycle (i.e., the reproductive cycle between two consecutive egg laying events, encompassing search for a host, blood feeding, oogenesis, and oviposition) and extrinsic incubation period (i.e., duration between infection onset in mosquitoes and the transmission, vertically in this study) on VT should vary depending on the mechanism. Trans-ovum transmission should be independent of the gonotrophic cycle as it is related to a tissue that is not altered by the oogenesis cycle, whereas transovarial transmission should be affected by oocyte development and release. The combined and separate impacts of gonotrophic cycle and incubation period for ZIKV VT in *Ae. aegypti* remain to be fully characterized.

In this study, we aimed at improving our understanding of ZIKV circulation by determining the impacts of gonotrophic cycle and extrinsic incubation period on VT in *Ae. aegypti*. To this end, we developed an experimental design that singled out progeny and larvae, allowing us to precisely determine: (i) oviposition rate, calculated as the proportion of blood-fed females that laid eggs; (ii) VT rate, defined as the proportion of infected females that transmit the virus to their progeny (at least one individual); (iii) filial infection rate, calculated as the proportion of infected larvae per progeny; and (iv) viral load per infected larvae, as determined by copy number of ZIKV genomic RNA (gRNA) [26]. The results shed light on the VT mechanism and help understand the balance between horizontal or vertical transmissions. By identifying the factors that influence ZIKV circulation in mosquitoes, our study will provide information to understand ZIKV persistence and inform vector control strategies. 

## 2. Results

### 2.1. Infection of Ovaries Increases with Extrinsic Incubation Time

Transovarial VT requires infection of mosquito female germinal tissues [21]. To quantify ovary infection, we collected ovary pairs from female *Ae. aegypti* at 3, 7, 10, and 14 days post oral infection (d.p.i.) with ZIKV and quantified gRNA copies by RT-qPCR. The different time points covered infection initiation up to complete infection of the whole mosquito body [10]. To orally infect mosquitoes, we used an infectious titer of 10^5^ particles forming unit (pfu) / mL, which is epidemiologically relevant as it corresponds to blood titers quantified in patients in Polynesia and the Americas [27,28]. We observed infection of ovaries as early as 3 d.p.i. with 80% of ovary pairs infected (Figure 1). Infection progressed to 100% of ovary pairs at 7, 10, and 14 d.p.i. Similarly, copies of ZIKV gRNA per infected ovary pairs gradually increased from an average of about 20,000 at 3 d.p.i. to an average of about 1.8 × 10^7^ at 14 d.p.i. (Figure 1). ZIKV gRNA copies were significantly lower at 3 d.p.i. than at 7 and 10 d.p.i., while gRNA level at 14 d.p.i. was significantly higher than at all earlier time points. The results support the early infection of *Ae. aegypti* ovaries following oral infection and indicate that virus load increases with extrinsic incubation period. 

### 2.2. Experimental Design to Test the Impacts of Gonotrophic Cycle and Extrinsic Incubation Time on VT

We designed an experiment to test the separate and interacting impacts of gonotrophic cycle and extrinsic incubation time on VT (Figure 2). Female mosquitoes were orally fed with 10^5^ pfu/mL of ZIKV as above, resulting in 100% infected mosquitoes (Figure 1). To minimize variation due to mosquito batch, the same batch of ZIKV-infected mosquitoes was divided into two cages. For each cage, we separated individual blood-fed mosquitoes in tubes prefilled with water and monitored oviposition at 3 d.p.i. The eggs from the first gonotrophic cycle were discarded as it has been shown that VT does not occur or is minimal in the first gonotrophic cycle for other arboviruses [29,30]. Mosquitoes that had oviposited and, thus, gone through their first gonotrophic cycle, were regrouped in their respective original cages. 

In one of the “sister” cages, mosquitoes were offered a non-infectious blood meal at 7 d.p.i., whereas the other cage was maintained on sugar feeding. To obtain a precise picture of VT, we separated individual blood-fed mosquitoes in tubes prefilled with water and collected eggs at 10 d.p.i. Each progeny was reared separately until fourth instar larvae. ZIKV was quantified in 10 single larvae randomly selected from each progeny. By separately analyzing blood-fed mosquitoes and larvae, we were able to record ovipositing females, proportion of infected larvae in a sub-sample of each progeny, and to quantify the viral load per individual infected larva. 

Mosquitoes that oviposited a second time had undergone a second gonotrophic cycle and were regrouped in the same cage. This cage and the one maintained on sugar feeding were then offered a non-infectious blood meal at 14 d.p.i. As previously, blood-fed mosquitoes from each cage were isolated in water-containing tubes. We collected the eggs at 17 d.p.i. and reared them until fourth instar larvae to quantify ZIKV in 10 single larvae per progeny. This experimental design quantified VT at 10 d.p.i. on the second gonotrophic cycle and at 17 d.p.i. on the second and third gonotrophic cycle. 

### 2.3. Oviposition Rate Moderately Increases with Incubation Time and Decreases with Additional Gonotrophic Cycle

Oviposition rate influences VT efficiency as absence of eggs prevents VT. We observed the highest oviposition rate for the second gonotrophic cycle at 17 d.p.i. with 79% of blood-fed mosquitoes laying eggs (Figure 3A). At the same extrinsic incubation time (17 d.p.i.), mosquitoes that had undergone an additional gonotrophic cycle (third) had a significantly lower oviposition rate at 54%. A shorter extrinsic incubation period at the second gonotrophic cycle reduced oviposition rate as compared to 17 d.p.i., although not significantly (*p*-value = 0.06, as indicated by Z-test). Our results indicate that oviposition rate is maximized by longer extrinsic incubation time and fewer gonotrophic cycles. 

### 2.4. VT Rate Drastically Increases with Incubation Time and Moderately Decreases with Gonotrophic Cycle

In each tested progeny, we quantified ZIKV in 10 randomly selected larvae. Although this experimental design may underestimate the VT rate by missing infected larvae in progeny with a low VT rate, the analysis of multiple progeny subsamples should provide an accurate observation. Strikingly, we observed that 100% of progeny were infected at 17 d.p.i. on the second gonotrophic cycle (Figure 3B). VT rate was significantly reduced to 76% when mosquitoes had undergone an additional gonotrophic cycle at 17 d.p.i. A larger reduction was caused by a shorter incubation time at the second gonotrophic cycle with only 21% of progeny infected. These results indicate that higher VT rate is mostly due to longer extrinsic incubation time, while processes associated with the gonotrophic cycle lower the VT rate.

### 2.5. Filial Infection Rate Increases with Incubation Time and Decreases with Gonotrophic Cycles

Calculation of filial infection rate was based on ZIKV detection in 10 single larvae randomly selected per progeny. Others quantify viruses in pools of larvae and calculate the minimum infection rate (MIR) based on the number of individuals per infected pool [7]. Both methods have pros and cons. However, our strategy also allowed us to calculate a complementary parameter: the exact viral load per infected larva (see below). 

Similarly to the VT rate, we observed the highest filial infection rate for the second gonotrophic cycle at 17 d.p.i. with 66% of infected larvae per progeny (Figure 3C). At 17 d.p.i., the filial infection rate was significantly reduced by an additional gonotrophic cycle (38%), while a shorter incubation lowered the infection rate even more at the second gonotrophic cycle to 8.5%. These results indicate that, similarly to the VT rate, a higher filial infection rate results from longer incubation time and fewer gonotrophic cycles. 

### 2.6. Viral Load per Infected Larva Increases with Incubation Time Only

We quantified the copy number of ZIKV gRNA per infected larva as an indication of viral load. Viral load was similar for larvae produced at 17 d.p.i. either from the second or third gonotrophic cycle (Figure 3D). The gRNA geometric mean was 816 for the second and 723 for the third gonotrophic cycle with values ranging from 103 to 10^8^ gRNA. A shorter incubation time (10 d.p.i.) significantly reduced the larval viral load at the second gonotrophic cycle to a geometric mean of 203 ranging from 106 to 1610 gRNA. The larval viral load most probably reflects the number of viruses that initially infect the eggs. These results indicate that the number of viruses transferred from mother to eggs increases solely with extrinsic incubation time.

## 3. Discussion

Most importantly, our results confirm the ability of ZIKV to be vertically transmitted up to fourth instar larvae in *Ae. aegypti* and reveal a surprisingly high VT efficiency. Our experimental design, which separated individual ovipositing mosquitoes and individual offspring from each progeny, provided a precise evaluation of VT including VT rate, filial infection rate [26], and the exact viral load per infected larva. VT efficiency was then calculated by multiplying the VT rate by the filial infection rate. With regards to the VT rate, we observed that 100% of progeny contained at least one infected larva in the most optimal condition (second gonotrophic cycle at 17 d.p.i.), whereas in the least optimal condition (second gonotrophic cycle at 10 d.p.i.), VT rate was 21%. Although we did not quantify VT at the first gonotrophic cycle at 3 d.p.i., the observed impact of incubation time suggests that VT rate would have been minimal. We found only one other study that separated progeny to assess dengue virus (DENV) VT rate in another *Aedes* species; the authors found a similar range of VT rates, from 12.5% to 94.7% [31]. In most other studies, larvae from different progeny were analyzed as a pool [12,13,14], preventing the calculation of VT rate. Our results provide the first evaluation of ZIKV VT rate in controlled conditions and reveal that the VT rate can be extremely high, while confirming that VT can largely vary. With regards to the filial infection rate, we reported 8.5–66% in the worst (second gonotrophic cycle at 10 d.p.i.) and best (second gonotrophic cycle at 17 d.p.i.) conditions, respectively. Usual pooled analyses underestimate the filial infection rate because it is calculated by making the conservative assumption that only one larva is infected per pool. For ZIKV and *Ae. aegypti*, three studies calculated a filial infection rate of 1.7–.3% by studying pools of 20 individuals [13], 2.13% by studying pools of 30 [12], and 5.5% by studying pools of 10 [14]. Actual filial infection rates for these three studies could be as high as 66%, 64%, and 55%, respectively, if all pooled individuals were infected. These numbers are definitely overestimates but are within the range of our results when calculated from a single larva from the same progeny. Another study quantified vertically transmitted ZIKV in single salivary glands and found a filial infection rate of 17% when eggs were collected at 5 d.p.i. [12], which is in line with the 21% we calculated in larvae from eggs collected at 10 d.p.i. Higher filial infection rate in salivary glands may be obtained when eggs are collected later, as observed for larvae from eggs we collected at 17 d.p.i. With regards to the viral load per infected larva, the best conditions (17 d.p.i. for either second or third gonotrophic cycle) averaged a bit lower than 1000 ZIKV gRNA per infected larva with a large variation among individual larvae. This is also the first time vertically transmitted viral load was calculated for individual larvae. The lowest infection levels we reported at the fourth larval instar may not persist through the mature phase and explain the lower filial infection rate observed in the mature than in the immature phase [7]. However, average and higher viral loads should be enough to infect the whole body of adults, including salivary glands. Vertically transmitted ZIKV infects salivary glands, enabling horizontal transmission to humans [14]. Overall, our precise results obtained by analyzing individual progeny and individual larvae indicate that VT efficiency in optimal conditions can be high (VT efficiency = VT rate × filial infection rate = 100% × 66% = 66%). 

Several lines of evidence suggest that ZIKV is vertically transmitted by transovarial infection. Infection of germinal cells is a prerequisite for transovarial infection. We detected ZIKV in ovaries as early as 3 d.p.i., as previously reported [12]. We quantified an exponential increase in viral loads until 14 d.p.i., suggesting active replication in ovaries. In support of germ cell infection by flaviviruses, particles of dengue virus (DENV), another mosquito-borne *Flavivirus*, are present in the germarium and oocytes of *Ae. aegypti* [25]. As compared to trans-ovum infection, which occurs through the oviduct, transovarial infection occurs in the germinal cells [21] and, thus, should be affected by oogenesis. We observed that an additional gonotrophic cycle reduces VT and filial infection rates. A similar impact for the gonotrophic cycle in *Ae. aegypti* was previously observed for DENV [32] and another mosquito-borne *Flavivirus*, yellow fever virus [33]. Eventually, during transovarial infection, longer incubation time should provide more time for the virus to infect more germinal cells. We observed that incubation time increased VT for ZIKV, as for DENV [25]. However, for the same incubation time, an additional gonotrophic cycle reduced the egg infection rate (as evaluated by VT rate and filial infection rate) but not infection intensity (as determined by viral load in individual larvae). This suggests that infection occurs during a stage of oogenesis that is renewed by subsequent cycles. In other words, oviposition shortens the actual incubation of the infected stage. The viral load received depends on the infection intensity in ovaries, which is identical for the same incubation time. During oogenesis, oocytes mature, are fertilized, and deposited, and another round of oocyte production is initiated from the germline cells. We thus propose that ZIKV infection occurs during oocyte maturation, although this hypothesis requires further testing. Strongly supporting transovarial infection of flaviviruses, VT for DENV was reported after egg surface sterilization [25]. Transovarial transmission should be more efficient than trans-ovum transmission [34] and could explain the high VT efficiency we observed.

Our results shed light on the balance between VT in mosquitoes and horizontal transmission to humans. We revealed that VT is higher for a longer extrinsic incubation time and lower number of gonotrophic cycles. These same conditions also favor oviposition rate, further increasing VT efficiency. Horizontal transmission occurs through blood feeding, which then stimulates oocyte maturation and search for an oviposition site [35]. The relationship between blood feeding and oviposition is central to balance horizontal transmission and VT; frequent blood feedings will favor horizontal transmission at the expense of VT and vis versa. *Aedes aegypti* is highly anthropophilic [36] and the limited availability of humans can reduce blood feeding. Alternatively, oviposition induces host seeking for blood feeding, which, when acquired, will initiate a subsequent gonotrophic cycle [37]. The alternation of gonotrophic cycles can then be driven by the availability of oviposition sites. Arid weather, for instance, should reduce *Ae. aegypti* preferred oviposition sites in artificial containers [38], delaying oviposition, and favoring VT. Mathematical models predict that in endemic settings, DENV would not persist by VT unless efficiency was higher than 20–30% [39,40]. Based on available data from pooled analyses, the authors of the model concluded that VT for DENV has a minimal impact on epidemiology. The model was based on *Ae. aegypti* biting habits and should partially apply to ZIKV. Our results showing VT higher than 30% for ZIKV suggest that VT can maintain the virus, at least when conditions limit availability of human hosts or oviposition sites. 

There is evidence that arboviruses can adapt to VT in mosquitoes. Laboratory studies showed that DENV can persist through several generations by VT [41,42,43]. DENV acquired by VT in *Ae. albopictus* females was then more efficiently vertically transmitted than DENV acquired by blood feeding [41]. It remains to be tested whether VT-adapted viruses conserve the ability to infect humans. Intriguingly, a meta-analysis found that VT efficiency was higher when flaviviruses were isolated from an arid climate [7], which, based on our results, corresponds to the optimal conditions for ZIKV VT. However, any adaptation to VT would have to be balanced with the potential fitness cost incurred to mosquito, as observed for slower growth of ZIKV vertically infected larvae [14]. The immune response in ovaries should also be taken into account. In our study, we used a low-passage ZIKV collected from humans (horizontally transmitted) in Brazil, which should not be adapted to VT. 

In conclusion, we deployed an original experimental design that allowed us to precisely calculate VT in *Ae. aegypti* for ZIKV and reveal VT efficiency higher than that quantified from pooled analysis. We further show that incubation time increases whereas successive gonotrophic cycle decreases VT, informing about the conditions that favor VT. Altogether, while horizontal transmission to humans remains the most epidemiologically relevant transmission mode, our study indicates that ZIKV VT may maintain the virus when conditions are not amenable for horizontal transmission. Consequently, it is advisable to maintain vector control when horizontal transmission (evidenced by outbreaks) is low in order to curb residual vertical transmission. 

## 4. Material and Methods

### 4.1. Mosquitoes

An *Ae. aegypti* mosquito colony was established from a thousand eggs collected in Singapore in 2010 and maintained in the laboratory. Eggs were hatched in tap water, larvae were fed a mix of fish food (TetraMin fish flakes, Melle, Germany) and liver powder (MP Biomedicals, Illkirch, France), and adults were held in rearing cages (Bioquip, Rancho Dominguez, CA, USA) supplemented with 10% sucrose and water. Adults were fed pig’s blood (Primary Industries Pte Ltd., Singapore) twice weekly to produce eggs for maintaining the colony. The insectary was held at 28 °C with 50% humidity on a 12:12 h dark:light cycle. 

### 4.2. Zika Virus

The strain BeH815744 was collected in the Paraiba state (northeast region), Brazil, in 2015 from a febrile non-pregnant woman with rashes. The virus was given from Pr. da Costa Vasconcelos, propagated in C6/36 (ATCC CRL-1660) cells and used after three passages. The virus stocks were titrated thrice by plaque assay in BHK-21 cells as previously described [10]. 

### 4.3. Oral Infection 

Three- to five-day-old female mosquitoes were sugar deprived for 24 h before offering a blood meal containing a 40% volume of washed erythrocytes from serum pathogen free (SPF) pig’s blood (SingHealth, Singapore), 5% of 10 mM ATP (Thermo Scientific, Waltham, MA, USA), 5% human serum (Sigma, St Louis, MO, USA) and 50% volume of virus in RPMI (Gibco, Waltham, MA, USA). The final blood viral titer was 10^5^ pfu/mL and was confirmed by a BHK-based plaque assay on blood samples collected before and after feeding. Mosquitoes were exposed to the artificial blood meal for one hour using a Hemotek membrane feeder system (Discovery Workshops, Accrington, UK) with a porcine intestine membrane. Fully engorged females were selected and maintained with free access to a 10% sugar solution and water in an incubation chamber with conditions similar to those used for insect rearing. 

### 4.4. Virus Quantification

Single ovary pairs or single larvae were homogenized in 350 µL of TRK lysis buffer (E.Z.N.A. Total RNA kit I (OMEGA Bio-Tek, Norcross, GA, USA)) using a bead Mill homogenizer (FastPrep-24, MP Biomedicals, Illkirch, France). Total RNA was extracted using the E.Z.N.A. Total RNA kit I (OMEGA Bio-Tek) and following the manufacturer’s instruction with 30 µL of DEPC-treated water for filter elution. ZIKV genomic RNA (gRNA) copies were quantified with a one-step RT-qPCR with SensiFAST SYBR No-ROX one-step kit (BioLine, London, UK). Primers, targeting the conserved region in the envelope, were: 5’- AGGACAGGCCTTGACTTTTC -3’ and 5’- TGTTCCAGTGTGGAGTTC -3’, as previously used [10]. The 10 µL reaction mix contained 400 nM of forward and reverse primers, and 3 µL of RNA extract. Quantification was conducted on a CFX96 Touch Real-Time PCR Detection System (Bio-Rad, Hercule, CA, USA). The thermal profile was 45 °C for 10 min, 95 °C for 1 min, and 40 cycles of 95 °C for 5 sec and 60 °C for 20 sec. 

An absolute standard curve was generated by amplifying fragments containing the qPCR targets using the qPCR forward primers tagged with a T7 promoter and the qPCR reverse primer. The fragment was reverse transcribed using a MegaScript T7 transcription kit (Ambion, Austin, TX, USA) and purified using the E.Z.N.A. Total RNA kit I. The total amount of RNA was quantified using a Nanodrop (Thermo Scientific) to estimate copy number based on an averaged base pair weight of 649 g/mole. Ten times serial dilutions were made and used to generate an absolute standard equation (Appendix A). In each subsequent RT-qPCR plate with samples, we quantified four standard aliquot dilutions to adjust for threshold variation between plates.

### 4.5. Experimental Design to Test the Impact of Gonotrophic Cycle and Extrinsic Incubation Period 

About 200 24 h starved 2- to 5-day-old female mosquitoes were offered a ZIKV infectious blood meal. Fully engorged mosquitoes were selected, and equally divided in two halves that were placed in separate 30 cm cubic cages (BioQuip, Rancho Dominguez, CA, USA). One day post-infectious blood feeding (d.p.i.), blood-fed mosquitoes were singled out in cylindrical vials of 2 cm diameter and 10 cm height. Vials were prefilled 1 cm high with distilled water and closed with a cotton plug. These eggs corresponding to the first gonotrophic cycle were collected at 3 d.p.i. and discarded. Only mosquitoes that oviposited were regrouped in their original cages. At 7 d.p.i., after 18 h of starvation, only one of the two cages was provided a non-infectious blood meal containing full SPF blood (SingHealth) for one hour using the Hemotek membrane feeder system with a porcine intestine membrane. Fully engorged mosquitoes were selected and kept in the cage, whereas non-fed mosquitoes were discarded. The other cage was identically starved but provided 10% sugar solution instead of a blood meal. At 8 d.p.i., mosquitoes that were fed a non-infectious blood meal were singled out as above in the cylindrical vials prefilled with distilled water. At 10 d.p.i., mosquitoes that oviposited were collected and kept in the same 30 cm cage. Eggs from each vial were reared in separated pans until the fourth instar larvae. At 14 d.p.i., after 18 h of starvation, the two cages were provided a non-infectious blood meal as above. Fully engorged females were selected and kept in their respective cages, whereas non-fed mosquitoes were discarded. At 15 d.p.i., mosquitoes from both cages were singled out in the vials prefilled with water. At 17 d.p.i., mosquitoes that oviposited were recorded and eggs were reared in separate pans until the fourth instar larvae. For each tested condition, we quantified ZIKV in 10 larvae randomly selected from one progeny. We analyzed 13–17 progeny per condition. During the experiment, mosquitoes were provided with 10% sugar solution when they were not offered a blood meal. Starvation consisted of removing the sugar solution. The experiment was repeated twice with different mosquito batches.

### 4.6. Statistics

Infection rate in the ovaries was calculated by dividing the number of infected ovaries over the total number of ovaries that were tested. VT rate was defined as the percentage of infected females that transmitted the virus to their progeny [21] and was calculated by dividing the number of progeny with at least one infected larva to the total number of females that oviposited. Filial infection rate was calculated by dividing the number of larvae with detectable levels of ZIKV gRNA out of 10 tested randomly selected larvae within progeny. Oviposition rate was calculated by dividing the number of female mosquitoes that oviposited over the total number of mosquitoes that were singled out in the vials prefilled with water. Differences in infection rates per progeny and ZIKV gRNA copies were evaluated using Tukey’s multiple comparison tests. After evaluating distribution normality with D’Agostino omnibus K² test, ZIKV gRNA copies were log-transformed before statistical analysis. Differences infection rates in ovaries and oviposition rates were evaluated using Z-test. All tests were performed with Prism v8.0.2 (GraphPad, San Diego, CA, USA), except for Z-tests that were conducted with Systat v13.1 (Systat Inc., San Jose, CA, USA).

## Figures and Tables

**Figure 1 pathogens-09-00366-f001:**
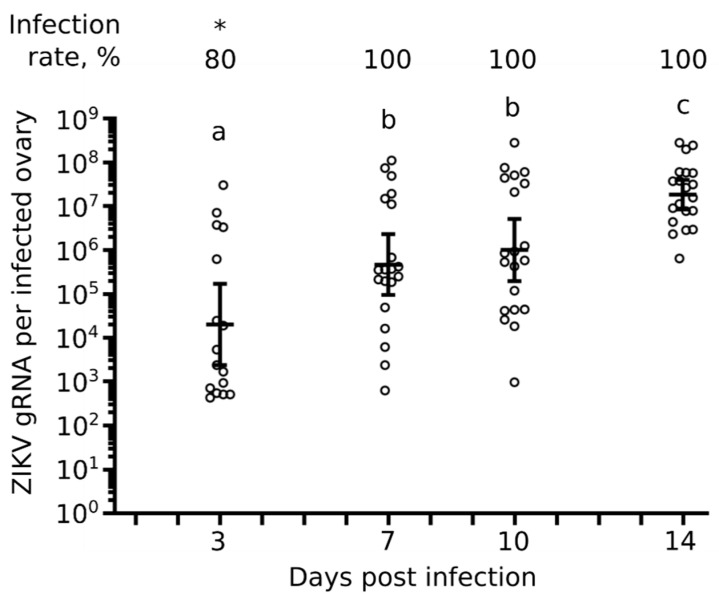
Kinetics of ovary infection. Zika virus (ZIKV) was quantified in single pairs of ovaries at 3, 7, 10, and 14 days post oral infection. Bars represent geometric means ± 95% CI. Each circle represents a pair of ovaries from one individual mosquito. N = 20 pair of ovaries per condition. * *p*-value < 0.05 from other conditions as calculated by Z-test. Different letters indicate significant difference (*p*-value < 0.05) as calculated by Tukey’s test.

**Figure 2 pathogens-09-00366-f002:**
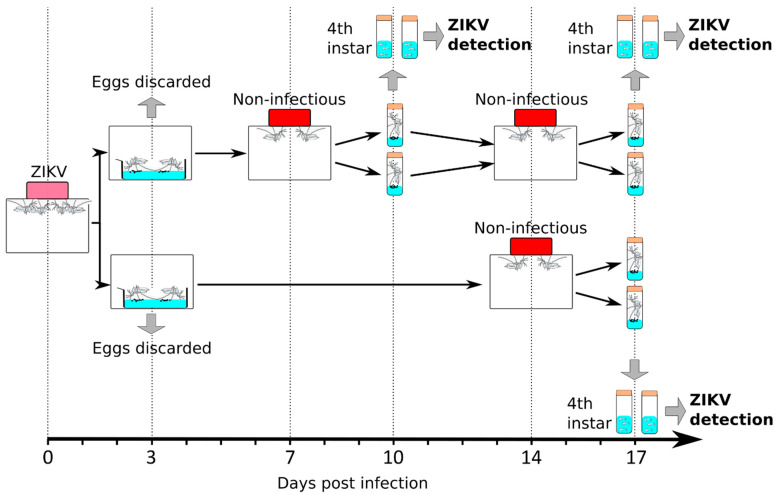
Experimental design to test the impact of gonotrophic cycle and time post infection on vertical transmission.

**Figure 3 pathogens-09-00366-f003:**
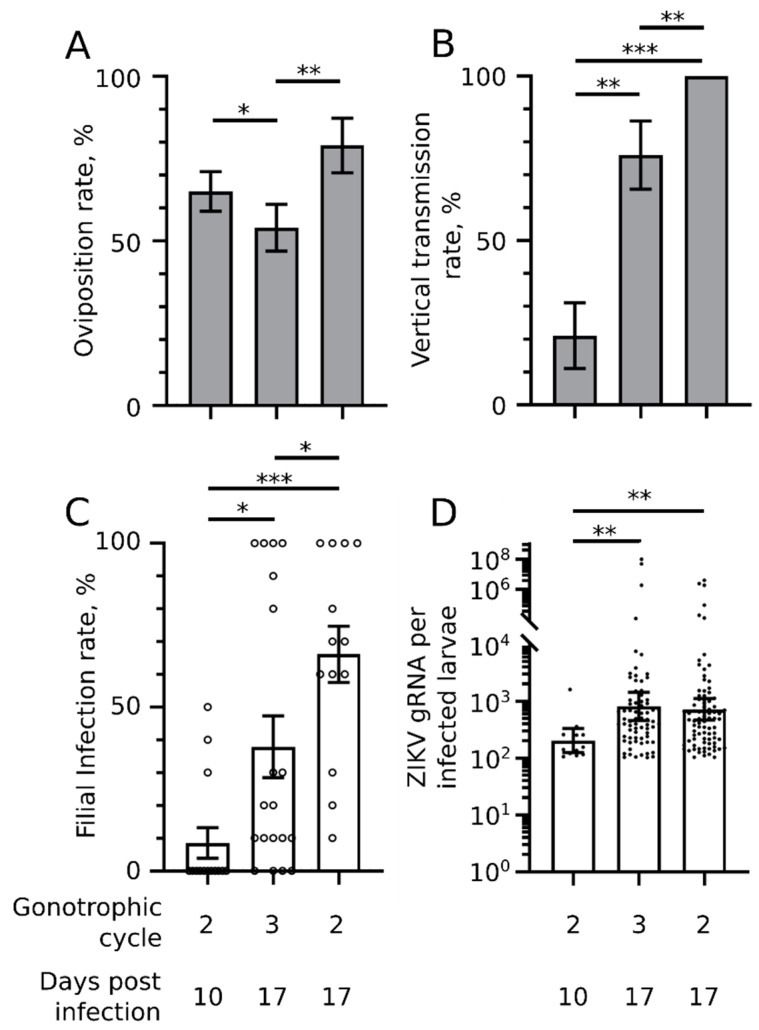
Impact of gonotrophic cycle and time post infection on vertical transmission. ZIKV was quantified in single fourth instar larvae on the second or third gonotrophic cycle at 10 or 17 days post oral infection. (**A**) Oviposition rate per condition. The number of mosquito females that laid eggs was divided by the total number of females that were blood fed. N blood-fed mosquitoes >24. (**B**) Vertical transmission rate. The number of progeny with at least one infected larva was divided by the total number of tested progeny. N progeny >14. (**A**,**B**) Bars indicate arithmetic means ± s.e. (**C**) Filial infection rate. Each circle represents the infection rate calculated from 10 single larvae randomly selected from all larvae from one infected mother (one progeny). Bars indicate arithmetic means ± s.e.m. (**D**) ZIKV genomic RNA (gRNA) copies per infected larvae. Each point represents one larva. Bars indicate geometric means ± 95% CI. (**C**,**D**) Larvae for each condition were collected from at least 14 progeny. * *p*-value < 0.05; **, *p*-value < 0.01; ***, *p*-value < 0.001, as calculated by Z-test (**A**,**B**) and Tukey’s test (**C**,**D**). Number of samples analyzed in each condition are detailed in Appendix A.

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
