# Peer review of "Highly Efficient Vertical Transmission for Zika Virus in Aedes aegypti after Long Extrinsic Incubation Time"

_pathogens, 2020, doi:10.3390/pathogens9050366_

Round 1

Reviewer 1 Report

The manuscript presented by Manuel et. al. targets importance of vertical transmission (VT) of Zika virus (ZIKV) for epizootic cycles and persistence of the virus in nature. Thus far, there has been conflicting data on the efficiency of VT and thus it´s influence virus transmission cycles in nature.

The presented data answer some to the  open questions regarding efficiency of VT in dependence on the gonotrophic cycle of the respective female. The authors show that the transmission of virus is most efficient in the second gonotrophic cycle, especially if this cycle is 17 instead of 7 days long. This argues, that despite the early spread of ZIKV within the mosquito, a certain time is needed for the virus to efficiently infect germline cells and that the level of ZIKV replication is the key factor for VT.

Overall, the data presented are sound and the information transported is important to understand the epizootics of ZIKV in nature especially the persistence of virus in the mosquito population under circumstances that are unfavourable for direct horizontal transmission cycles such as the absence of a blood host and the shortage of breeding sites for egg deposition.

I have a few suggestions that could be targeted by the authors:

  • In general, all experiments were carried out with great care. However, I was wondering why there was not attempt to test the first egg batch at 3 days p.i. although 80% if the ovaries were found infected. I can follow the argument that the VT was tested low at the first gonotrophic cycle for other arboviruses. Nevertheless, is it certain that the first egg batch in Zika infected mosquitoes would be negative because Yellow fever and West Nile are not transmitted to eggs at this time?
  • Could you define gonotrophic cycle and extrinsic incubation time in the introduction to make the text easier to understand for non-experts?

  • It would be interesting to know the real number of mosquitos used at each step of the protocol for example in addition to percentage values. How many of the 200 mosquitos initially put into the experiment did fee, lay eggs and so on.

  • Although maybe a bit too farfetched, I would be interested in the author’s views of mosquito immunity and the efficiency of VT. It is known that some viruses induce next to siRNA also piRNA pathways, which are known to be most efficient in germline cells. Would this piRNAi have an impact on VT?

Author Response

Answers to reviewers in blue. Lines refer to the changes in the track-change version of the revised manuscript.

Overall, we are in depth to the reviewers for highlighting the quality of the experimentation and the soundness of the data. We have addressed each reviewers’ comments and modified the text accordingly;

The manuscript presented by Manuel et. al. targets importance of vertical transmission (VT) of Zika virus (ZIKV) for epizootic cycles and persistence of the virus in nature. Thus far, there has been conflicting data on the efficiency of VT and thus it´s influence virus transmission cycles in nature.

The presented data answer some to the  open questions regarding efficiency of VT in dependence on the gonotrophic cycle of the respective female. The authors show that the transmission of virus is most efficient in the second gonotrophic cycle, especially if this cycle is 17 instead of 7 days long. This argues, that despite the early spread of ZIKV within the mosquito, a certain time is needed for the virus to efficiently infect germline cells and that the level of ZIKV replication is the key factor for VT.

Overall, the data presented are sound and the information transported is important to understand the epizootics of ZIKV in nature especially the persistence of virus in the mosquito population under circumstances that are unfavourable for direct horizontal transmission cycles such as the absence of a blood host and the shortage of breeding sites for egg deposition.

I have a few suggestions that could be targeted by the authors:

In general, all experiments were carried out with great care. However, I was wondering why there was not attempt to test the first egg batch at 3 days p.i. although 80% if the ovaries were found infected. I can follow the argument that the VT was tested low at the first gonotrophic cycle for other arboviruses. Nevertheless, is it certain that the first egg batch in Zika infected mosquitoes would be negative because Yellow fever and West Nile are not transmitted to eggs at this time?

We now discuss the VT that would have been observed had we measured it at the 1st gonotrophic cycle: “Although we did not quantify VT at the 1st gonotrophic cycle at 3 d.p.i., the observed impact of incubation time suggests that VT rate would have been minimal.” (l. 200-201)

  • Could you define gonotrophic cycle and extrinsic incubation time in the introduction to make the text easier to understand for non-experts?

We thank reviewers for suggesting clarification and have now added the definition of gonotrophic cycle when we first mention this term: “(i.e., the reproductive cycle between two consecutive egglaying events, encompassing search for a host, blood feeding, oogenesis and oviposition) (l. 65-67).

Following on this comment, we now also defined extrinsic incubation time in the first occurrence (l. 67-68).

  • It would be interesting to know the real number of mosquitos used at each step of the protocol for example in addition to percentage values. How many of the 200 mosquitos initially put into the experiment did fee, lay eggs and so on.

We added a supplementary table with these numbers (Table S1).

  • Although maybe a bit too farfetched, I would be interested in the author’s views of mosquito immunity and the efficiency of VT. It is known that some viruses induce next to siRNA also piRNA pathways, which are known to be most efficient in germline cells. Would this piRNAi have an impact on VT?

In the revised discussion, we now mention the potential role of ovary immunity (l. 279). However, we feel that giving more details would require a more in-depth study on ovary immunity.

Reviewer 2 Report

In this manuscript, the authors sought to assess the impacts of viral incubation time and gonotrophic cycling on the efficiency of vertical transmission of Zika virus in Aedes aegypti mosquitoes. The authors infect mosquitoes via infected blood meal and assess viral load (via genome copies) in the mosquito ovaries at different timepoints post infection. Using another set of infected mosquitoes, the authors assessed the efficiency of vertical transmission (by looking at whether progeny were infected) under different egg laying parameters such as time between initial infection and oviposition and the number of ovipositions that occurred. Overall, the authors conclude that longer time between infection and oviposition and fewer gonotrophic cycles resulted in increased vertical transmission efficiency.

Overall, study design was sound, and the methods employed were suitable to assess the authors' question. Results are displayed and explained clearly. The conclusions were thoroughly discussed; however, the discussion is too long for the size of this study. Please shorten/condense the discussion.

While well designed and executed, the significance of this study is not readily apparent. As a field, we already know that increased extrinsic incubation time increases vertical transmission of flaviviruses. It is also already known that Ae. aegypti mosquitoes can vertically transmit Zika virus. Please elaborate in the introduction and/or discussion on the significance study. Do your results help inform mosquito control measures, could this information be used to assess transmission risk, etc?

Author Response

Answers to reviewers in blue. Lines refer to the changes in the track-change version of the revised manuscript.

Overall, we are in depth to the reviewers for highlighting the quality of the experimentation and the soundness of the data. We have addressed each reviewers’ comments and modified the text accordingly;

In this manuscript, the authors sought to assess the impacts of viral incubation time and gonotrophic cycling on the efficiency of vertical transmission of Zika virus in Aedes aegyptimosquitoes. The authors infect mosquitoes via infected blood meal and assess viral load (via genome copies) in the mosquito ovaries at different timepoints post infection. Using another set of infected mosquitoes, the authors assessed the efficiency of vertical transmission (by looking at whether progeny were infected) under different egg laying parameters such as time between initial infection and oviposition and the number of ovipositions that occurred. Overall, the authors conclude that longer time between infection and oviposition and fewer gonotrophic cycles resulted in increased vertical transmission efficiency.

Overall, study design was sound, and the methods employed were suitable to assess the authors' question. Results are displayed and explained clearly. The conclusions were thoroughly discussed; however, the discussion is too long for the size of this study. Please shorten/condense the discussion.

We have tried to shorten the discussion while maintaining the thoughness that the reviewer acknowledged. From the discussion, we removed:        
1. the introductory paragraph, which presented a summary of the introduction

2. the definitions of VT rate and filial infection rate, which definitions were given in the introduction.

3. Parts of sentences that were not absolutely necessary.

While well designed and executed, the significance of this study is not readily apparent. As a field, we already know that increased extrinsic incubation time increases vertical transmission of flaviviruses. It is also already known that Ae. aegypti mosquitoes can vertically transmit Zika virus. Please elaborate in the introduction and/or discussion on the significance study. Do your results help inform mosquito control measures, could this information be used to assess transmission risk, etc?

We thank the reviewer for requesting us to extract all the consequences of our results.
We now added in the introduction that “Identification of the mode of transmission to target will help curb residual circulation.” (l. 38) and in the discussion that “it is advisable to maintain vector control when horizontal transmission (evidenced by outbreaks) is low in order to curb residual vertical transmission.” (l. 287-288).